# Research on the Cooperative Target State Estimation and Tracking Optimization Method of Multi-UUV

**DOI:** 10.3390/s23187865

**Published:** 2023-09-13

**Authors:** Tao Chen, Qi Qi

**Affiliations:** 1College of Intelligent Systems Science and Engineering, Harbin Engineering University, Harbin 150001, China; chentao_7777@163.com; 2Qingdao Innovation and Development Base, Harbin Engineering University, Qingdao 266000, China

**Keywords:** UUV, cooperative target tracking, fusion estimation, tracking optimization, particle swarm optimization, bearing-only, extended Kalman Filter, interacting multiple model

## Abstract

This work studied two sub-problems of the cooperative state estimation and cooperative optimization of tracking paths in multiple unmanned underwater vehicle (multi-UUV) cooperative target tracking. The mathematical model of each component of the multi-UUV cooperative target tracking system was established. According to the target bearing-only information obtained by each unmanned underwater vehicle’s (UUV) detection, the extended Kalman filter algorithm based on interacting with multiple model bearing-only data was used to estimate the target state in a distributed way, and the federal fusion algorithm was used to fuse the estimated results of each UUV. The fused target state was predicted, and, based on the predicted target state, to achieve the persistent tracking of the target, the particle swarm optimization algorithm was used for the online collaborative optimization of the UUV tracking path. The simulation results showed that the multi-UUV distributed fusion filtering algorithm could obtain a better target state estimation effect, and the online path collaborative optimization method based on the prediction of the target state could achieve persistent target tracking.

## 1. Introduction

Target tracking is a hot research direction in the field of UUV applications. In the underwater environment, the UUV mainly depends on its own passive sonar to judge the relative orientation information of a target according to the noise radiation of the target [1]. Under the background of a complex underwater environment, less detection information, and uncertain target motion, it is often very difficult to persistently track a target by only relying on a single UUV. However, multi-UUV cooperative target tracking makes up for these shortcomings. Multi-UUV cooperatively estimates the target state and fuses the estimated results to achieve a high-precision estimation of the target’s motion state, which greatly reduces the target loss probability in the tracking process. While multi-UUV cooperative target tracking brings these advantages, it also brings two major challenges: one is how to fuse the estimation results of multi-UUVs; and the second is how to optimize the tracking path online according to the estimation results to achieve better detection [2].

In reference [3], a path-planning strategy combining the Lyapunov principle and a collision avoidance potential function was proposed to solve the problem of unmanned aerial vehicle (UAV) ground target tracking in the presence of obstacles and motion threats, which achieves collision avoidance while target tracking. Reference [4] used an interactive multi-model algorithm based on a second-order Markov chain to track maneuvering targets and verified the effectiveness of this algorithm compared to other multi-model algorithms. Reference [5] proposed a multi-UAV path-planning algorithm based on the spatial refined voting mechanism (SRVM) and particle swarm optimization (PSO). Reference [6] mainly studied a multiple autonomous underwater vehicle (AUV) system for cooperative tracking. Reference [7] analyzed the research status of cooperative estimation in the navigation field and presented a novel methodology of a fading cubature Kalman filter (CKF) with an augmented mechanism to address the problems involved in tightly coupled inertial navigation system/celestial navigation system (INS/CNS) integration for hypersonic vehicle (HV) navigation. Finally, it was verified that this method has a good performance in HV navigation with tightly coupled INS/CNS integration.

In this paper, a distributed target state estimation algorithm based on IMM-EKF is proposed to solve the problem of multi-UUV cooperative target tracking. Each UUV performs target state estimation in a distributed manner and the estimation results of each UUV are fused through a federal fusion structure. A collaborative path optimization method based on particle swarm optimization (PSO) is proposed to solve the problem of UUV tracking path optimization. One-step prediction is performed based on the fusion filtering results, the predicted target state is used as the path optimization input to optimize the UUV tracking path online in real time, and the tracking performance is improved.

## 2. Problem Description and Modeling

### 2.1. Problem Description

Firstly, the definition of the multi-UUV cooperative target tracking problem is given as follows:

**Definition** **1.**
*(Multi-UUV cooperative target tracking problem) assumes that there are multiple UUVs and moving targets with limited prior knowledge in the space. Each UUV carries sensors and can detect target information, which requires each UUV to cooperatively complete the target state estimation according to the detection information. According to the target state estimation result and each UUV space constraint, the tracking path is adjusted to complete the target tracking task.*


Definition 1 can be modeled by mathematical formulation. The first is the mathematical model of fusion estimation:(1)$$\begin{array}{c} \max(\lambda (P(X_{t}[k]|Z_{i}[k],X_{t}[k-1])))\\ s.t.\;X_{t}[k]=f(X_{t}[k-1])+Q[k]\\ \;\;\;\;Z_{i}[k]=h_{i}(X_{t}[k],X_{u,i}[k])+\omega [k],i=1\cdot \cdot \cdot N \end{array},$$
where P(⋅) is the conditional probability description of multi-UUV cooperative target state fusion estimation, λ(⋅) is the corresponding scalar function. $X_{t}[k]$ is the target state variable at time k, and $X_{t}[k-1]$ is the target state variable at time k−1. $X_{u,i}[k]$ is the state variable of the i-th UUV at time k. f(⋅) is the target state transition function. $Z_{i}[k]$ represents the detection information of the i-th UUV at time k. $h_{i}(\cdot )$ is the nonlinear detection function of the i-th UUV. Q[k] represents the process noise of the target model and ω[k] is the detection noise of the sonar. N is the total number of UUVs executing tracking tasks.

The result of the target state fusion estimation will be used as the basis for multi-UUV tracking path optimization. The tracking optimization problem can be described as:(2)$$X_{com}[k]=\max(\sum_{i=1}^{N}J_{sen}(X_{t}[k],X_{u,i}[k])),$$
where $X_{com}[k]$ is the instruction sequence of the UUV at time k and $J_{sen}(\cdot )$ is the efficiency function of sonar detection, which is described as follows:(3)$$J_{sen}(X_{t}[k],X_{u,i}[k])=\left\{\begin{array}{c} 1,\\ 0, \end{array}\begin{array}{c} The\;target\;is\;not\;lost\\ The\;target\;has\;been\;lost \end{array}\right..$$

The above mathematical model is understood as follows. The purpose of multi-UUV cooperative target state fusion estimation is, under the condition that the target state $X_{t}[k-1]$ at time k−1 and multi-UUV detection information $Z_{i}[k]$ at time k are known, that the estimation result $X_{t}[k]$ of the target state at time k is optimal. The control instruction sequence of a multi-UUV solved by the optimization algorithm should strive to ensure that more UUVs meet the requirements of persistent tracking. In other words, on the premise that the target is detected by as many UUVs as possible, the optimal control sequence for a multi-UUV is solved.

According to Definition 1, the multi-UUV cooperative target tracking problem in this paper can be described as follows:

First of all, the tracking plane is two-dimensional and the target is mobile. In the process of motion, the target may change its motion mode at any time. The detection method is bearing-only detection. The UUV carries passive sonar, which can detect the relative bearing information of the target. The tracking mode is active tracking, and the UUV will adjust the tracking path online according to the target state fusion estimation results during the tracking process. The requirement of the tracking task is persistent tracking, that is, to keep the target within the detection range of each UUV at all times. The schematic diagram of multi-UUV cooperative target tracking is shown in Figure 1.

### 2.2. Dynamic and Kinematic Models of UUV

Assuming that the UUV only moves in a two-dimensional plane, the following dynamic equation can be established for the UUV [8]:(4)$$\left\{\begin{array}{l} \overset{•}{u}=\frac{F}{m_{11}}-\frac{d_{11}}{m_{11}}u+\frac{m_{22}}{m_{11}}vr\\ \overset{•}{v}=-\frac{m_{11}ur}{m_{22}}-\frac{d_{22}}{m_{22}}v\\ \overset{•}{r}=\frac{T}{m_{33}}-\frac{d_{33}}{m_{33}}r+\frac{m_{11}-m_{22}}{m_{33}}uv \end{array}\right.,$$
where $m_{11}=m-X_{u}$, $m_{22}=m-X_{v}$, $m_{33}=I_{z}-N_{r}$, $d_{11}=-X_{u}$, $d_{22}=-Y_{v}$, $d_{33}=-N_{r}$, u, and v are the longitudinal and transverse velocities of the UUV, respectively; r is the bow angular velocity of the UUV; m is the mass of the UUV; $I_{z}$ is the fixed torque of the UUV on the shaft; Z is the longitudinal thrust of the UUV propeller; T is the UUV bow torque; and $X_{*}$, $Y_{*}$, and $N_{*}$ are the viscous hydrodynamic coefficients.

The UUV dynamic model describes the relationship between force, torque, and UUV velocity. In order to describe the relationship between the UUV’s velocity and position, and between the bow’s angular velocity and heading, a kinematic model of the UUV is established as follows:(5)$$\left\{\begin{array}{l} \dot{x}=u\cos\psi -v\sin\psi \\ \dot{y}=u\sin\psi +v\cos\psi \\ \dot{\psi }=r \end{array}\right.,$$
where x and y are the positions of the UUV in the geodetic coordinate system and ψ is the heading of the UUV.

### 2.3. Target Motion Model

Assume that the target only moves in the two-dimensional plane, and its state variable is $X={\left[\begin{array}{cccc} x_{t} & \overset{•}{x_{t}} & y_{t} & \overset{•}{y_{t}} \end{array}\right]}^{T}$. The mathematical description of the target motion is:(6)$$X_{k+1}=FX_{k}+GQ_{k},$$
where F is the state transition matrix, G is the control matrix, and $Q_{k}$ is the Gaussian white noise with a mean of 0 and covariance of $\sigma^{2}$. The state transition matrix includes the uniform motion model and uniform turning model. Its matrix representation is as follows:$$F_{CV}=\left[\begin{array}{cccc} 1 & 0 & T & 0\\ 0 & 1 & 0 & T\\ 0 & 0 & 1 & 0\\ 0 & 0 & 0 & 1 \end{array}\right],\;G_{CV}=\left[\begin{array}{cc} \frac{T^{2}}{2} & 0\\ 0 & \frac{T^{2}}{2}\\ T & 0\\ 0 & T \end{array}\right],\;F_{CT}=\left[\begin{array}{cccc} 1 & \frac{\sin\omega T}{\omega } & 0 & \frac{1-\cos\omega T}{\omega }\\ 0 & \cos\omega T & 0 & -\sin\omega T\\ 0 & \frac{1-\cos\omega T}{\omega } & 1 & \frac{\sin\omega T}{\omega }\\ 0 & \sin\omega T & 0 & \cos\omega T \end{array}\right],\;G_{CT}=\left[\begin{array}{cc} \frac{T^{2}}{2} & 0\\ 0 & \frac{T^{2}}{2}\\ T & 0\\ 0 & T \end{array}\right]$$
where T is the sampling time and ω is the turning angular velocity.

### 2.4. Bearing-Only Detection Model

It is assumed that the UUV carries passive sonar, and the detection mode is bearing-only detection. The detection information does not consider other detection information except the relative azimuth, and the detection plane is in the two-dimensional plane, without considering the depth factor. The detection angle of the UUV’s passive sonar at time k can be described as follows:(7)$$Z[k]=\arctan\frac{x_{t}-x_{u}}{y_{t}-x_{u}}+\omega [k],$$

Among them, $x_{t}$ and $y_{t}$ are the target positions at k, $x_{u}$ and $y_{u}$ are the UUV positions at k, and ω[k] is the detection noise of the passive sonar at k.

## 3. Fusion Estimation and Tracking Optimization Solving Framework

### 3.1. Tracking Optimization Approach Based on Target Prediction States

According to the different target states used for UUVs, the tracking optimization approach includes two aspects: an approach based on current target states and an approach based on target prediction states. The tracking optimization approach based on target prediction states is adopted in this paper.

In the process of target tracking, if only the target’s current states are considered, there is a great risk of target loss. As shown in Figure 2, assuming that path optimization is performed every 3 s, the target may be out of the detection range of the UUV after two seconds, and the UUV still executes the instructions obtained based on $X_{t}[k]$.

According to the above problem, some studies have addressed this predicament through a diverse range of techniques, such as random weighting, Sage windowing, and adaptive factor adjustments [9,10]. In this paper, another solution is reached on the basis of the current target state fusion estimation results to predict the state of the target at the future time based on the target prediction state to optimize the tracking path of the UUV, which is shown in Figure 3. $X_{t}^{-}[k+1]$ is the target prediction state on the k+1 moment based on the estimation results $X_{t}^{}[k]$. In addition, the tracking structure based on the target predicted state is shown in Figure 4, which greatly reduces the probability of target loss in the tracking process.

### 3.2. Finite Centralized Distributed Solving Framework

There are three commonly used control structures in multi-UUV cooperative control systems: centralized, distributed, and hybrid [11]. In the centralized architecture, there is a central node, and all UUV nodes communicate with the central node directly. In the distributed control structure, there is no central node, and UUVs communicate with each other. In the hybrid structure, there is a manual control center, and all UUVs communicate with the control center and can also communicate with each other.

At present, the multi-UUV system is not advanced enough to perform tasks completely autonomously without central intervention, so the existence of a central control node is inevitable. At the same time, it is hoped that each UUV has a certain autonomous ability, and can communicate and cooperate equally. Therefore, the hybrid control structure is an ideal control structure applied to multiple UUV systems. Combined with the above closed-loop tracking task process based on predicting the target state, a finite-set distributed solution framework for solving the multi-UUV cooperative target tracking problem is given, as shown in Figure 5.

By solving the framework shown in Figure 5, it can be seen that each is a closed loop. Multiple UUVs communicate in the task execution process through collaborative target state estimation and tracking path optimization, and each UUV in the task process is linked, in both the artificial control of monitoring and management. In this solving process, there are two main points of collaboration between UUVs:

(1)The target state information is exchanged between the UUVs to complete the fusion estimation of the target state;(2)According to the respective position information and target estimation information fusion between the UUVs, the tracking path optimization is completed.

The two collaborations between UUVs also reflect the two keyword problems of solving the multi-UUV cooperative target tracking problem, the multi-UUV cooperative target state fusion estimation problem, and the multi-UUV tracking path cooperative optimization problem.

## 4. Distributed Cooperative Target State Fusion Estimation Method

### 4.1. Structure of Distributed Fusion Estimation Based on IMM-EKF and Federated Fusion

In the process of tracking a target, the sonar detects the orientation information of the target. According to the bearing-only detection model of UUV sonar, the observation equation is nonlinear, so the nonlinear filtering method should be used to estimate the target state. Common nonlinear filtering methods are: extended Kalman filter, unscented Kalman filter, and particle filter [12]. Considering the practical application of UUV target tracking and the advantages and disadvantages of the various filtering methods, the extended Kalman filter algorithm with a simple algorithm, strong engineering implementation ability, and better real-time performance is selected in this paper [13,14].

In the process of target tracking, due to the variable motion of targets, it is often difficult for a single model to match the real motion of a target. The interacting multiple model (IMM) is a good and computationally efficient method for dealing with the model-matching problem of maneuvering target tracking. Reference [15] applied the IMM algorithm to the vehicle state estimation problem and achieved good results. Combined with the extended Kalman filter algorithm, this paper uses the extended Kalman filter algorithm based on the interacting multiple model (IMM-EKF) to estimate the target state.

The choice of fusion structure also has an important impact on the problem of multi-UUV cooperative estimation. There are three commonly used fusion structures: a distributed structure, a centralized structure, and a hierarchical structure [16]. According to the previous problem description, the task requirement of the multi-UUV cooperative target tracking studied in this paper is persistent tracking, that is, ensuring that the target is within the detection range of each UUV at all times. Assuming that the detection range of the UUVs is equal to the communication range, communication between the UUVs must be possible on the premise of satisfying the persistent tracking. Because all UUVs can communicate with each other, each UUV can make use of the detection information of all UUVs when performing fusion estimation, and as long as each UUV uses the same fusion rule, the consistency of the estimation results is not considered.

In summary, the structure of distributed fusion estimation based on IMM-EKF and federated fusion is shown in Figure 6.

### 4.2. IMM-EKF Estimation Method

When the target state estimation is conducted and the distributed target state estimation method is used, each UUV according to itself detects the target information independently of the target state estimation based on the IMM-EFK algorithm, which is passed through mutual communication between the estimated results and estimated covariance, finally according to their own estimation information and the other UUV estimation information fusion estimation. The specific steps of the IMM-EKF algorithm are as follows.

(1)Input interaction

We first calculate the mixing probability from model i to model j as follows:(8)$$u_{ij}(k-1|k-1)=\sum_{i=1}^{r}u_{i}(k-1)/\bar{c_{j}},$$
where the normalization constant $\bar{c_{j}}$ is calculated by the following equation:(9)$$\bar{c_{j}}=\sum_{i=1}^{r}p_{ij}u_{i}(k-1),$$
where r represents the number of models in the target motion model set, $p_{ij}$ represents the transition probability from model i to model j, and $u_{i}(k-1)$ represents the probability of model i at time k−1.

The mixed-state estimates and mixed covariance estimates for model j can be further obtained:(10)$$\hat{X_{0j}}(k-1|k-1)=\sum_{i=1}^{r}\hat{X_{i}}(k-1|k-1)u_{i}(k-1|k-1),$$
(11)$$\begin{array}{c} P_{0j}(k-1|k-1)=\sum_{i=1}^{r}u_{ij}(k-1|k-1)\{P_{i}(k-1|k-1)+\\ \;[\hat{X_{i}}(k-1|k-1)-\hat{X_{0j}}(k-1|k-1)]•\\ \;[\hat{X_{i}}(k-1|k-1)-\hat{X_{0j}}(k-1|k-1)]T\} \end{array},$$
where j=1,2,⋅⋅⋅,r,$\hat{X_{i}}(k-1|k-1)$, and $P_{i}(k-1|k-1)$ are the state and covariance estimates of model j at time k−1, respectively.

(2)Extended Kalman filter

Taking $\hat{X_{i}}(k-1|k-1)$, $P_{i}(k-1|k-1)$, and probe information Z as the inputs, the extended Kalman filter is performed.

State prediction:(12)$$\hat{X_{j}}(k|k-1)=F_{j}(k-1)\hat{X_{0j}}(k-1|k-1),$$

Error covariance prediction:(13)$$P_{j}(k|k-1)=F_{j}P_{0j}(k-1|k-1)F_{j}^{T}+G_{j}Q_{j}G_{j}^{T},$$

The Jacobian matrix is calculated. Under the bearing-only detection model, the target observation equation is nonlinear, and the Jacobian matrix needs to be solved to linearize the approximation. The Jacobian matrix is solved as follows:(14)$$H=\left[\frac{\partial Z}{\partial x_{t}},\frac{\partial Z}{\partial \overset{•}{x_{t}}},\frac{\partial Z}{\partial y_{t}},\frac{\partial Z}{\partial \overset{•}{y_{t}}},\frac{\partial Z}{\partial \overset{••}{x_{t}}},\frac{\partial Z}{\partial \overset{••}{y_{t}}}\right],$$
calculate the Kalman gain:(15)$$K_{j}(k)=P_{j}(k|k-1)H^{T}{\left[HP_{j}(k|k-1)H^{T}+R\right]}^{-1},$$
status updates:(16)$$\hat{X_{j}}(k|k)=\hat{X_{j}}(k|k-1)+K_{j}(k)•\left[Z(k)-arctan\frac{x_{t}^{-}-x_{u}}{y_{t}^{-}-y_{u}}\right],$$
covariance update:(17)$$P_{j}(k|k)=\left[I-K_{j}(k)H(k)\right]P_{j}(k|k-1),$$
where $F_{j}(k-1)$ is the state transition matrix in the target motion model set, $G_{j}$ is the control matrix, $Q_{j}$ is the process noise, R is the detection noise, $x_{t}$, $\overset{•}{x_{t}}$, $y_{t}$, $\overset{•}{y_{t}}$, $\overset{••}{x_{t}}$, and $\overset{••}{y_{t}}$ are the state variables of the target, $x_{t}^{-}$ and $y_{t}^{-}$ are the target predicted positions obtained according to the target model, and $x_{u}$ and $y_{u}$ are the UUV positions.

(3)Model probability update

Calculate the likelihood function for model j:(18)$$\Lambda_{j}(k)=\frac{1}{{\left(2\pi \right)}^{n/2}|S_{j}(k){|}^{1/2}}•\exp\left\{-\frac{1}{2}v_{j}^{T}S_{j}^{-1}(k)v_{j}\right\},$$
among them:(19)$$v_{j}(k)=Z(k)-H(k)\hat{X_{j}}(k|k-1),$$
(20)$$S_{j}(k)=H(k)P_{j}(k|k-1)H{\left(k\right)}^{T}+R(k),$$
the probability of model j is:(21)$$u_{j}(k)=\Lambda_{j}(k)\bar{c_{j}}/c,$$
where the normalization constant is:(22)$$c=\sum_{j=1}^{r}\Lambda_{j}(k)\bar{c_{j}},$$

(4)Output interaction

Based on the model probabilities, the final state estimate and covariance are obtained using the weighted summation of the estimated results for each filter as follows:(23)$$\hat{X}(k|k)=\sum_{j=1}^{r}\hat{X_{j}}(k|k)u_{j}(k),$$
(24)$$\begin{array}{c} P(k|k)=\sum_{j=1}^{r}u_{j}(k)\left\{P_{j}(k|k)+\right.\\ \;\left[\hat{X_{j}}(k|k)-\hat{X}(k|k)\right]•\\ \;\left.{\left[\hat{X_{j}}(k|k)-\hat{X}(k|k)\right]}^{T}\right\} \end{array}.$$

### 4.3. Federated Fusion Estimation Method

In order to improve the accuracy of the target state estimation with multi-UUV, this paper adopts the collaborative estimation method of federal fusion. After each UUV completes the state estimation, it transmits the obtained target state and covariance information to all the other UUVs by broadcasting. Each UUV performs the weighted fusion of the received target state information and the target state information obtained via its own estimation according to the corresponding covariance. The specific fusion rules are as follows:(25)$$\hat{X_{g}}=P_{g}\sum_{i=1}^{N}P_{ii}^{-1}\hat{X_{i}},$$
(26)$$P_{g}={\left(\sum_{i=1}^{N}P_{ii}^{-1}\right)}^{-1},$$
where N denotes the number of sub-filters, $\hat{x_{i}}$ denotes the local state estimate of the i-th sub-filter, and $p_{ii}$ denotes the corresponding covariance. $\hat{x_{g}}$ is the final target state estimation result after fusion.

Finally, the distributed cooperative target state fusion estimation algorithm based on IMM-EKF can be described as follows:

Finally, the distributed cooperative target state fusion estimation algorithm based on IMM-EKF can be described in Table 1:

## 5. Tracking Path Cooperative Optimization Method Based on PSO

### 5.1. Optimization Function Design for Persistent Target Tracking Task

The goal of the target tracking task is not to lose the target, that is, to keep the target in the detection range of the UUV sensor, and on this premise, to optimize the tracking path of the UUV to achieve the better detection of the target and reduce the target state estimation error. Firstly, according to the task requirements and practical engineering applications, the objective function of the optimization problem, namely, the fitness function of the particle swarm optimization algorithm, can be designed as follows:(27)$$f=J_{pers}+J_{cros}+J_{safe},$$

The fitness function is composed of three parts, and each part is a penalty function. In the process of seeking the optimal solution in line with the optimization objective, the smaller the fitness function value of the particle position, the smaller the penalty value of the position, indicating that the position is better. The detailed description of the three-part penalty function is as follows:

(1)Persistent tracking penalty function


(28)
$$J_{pers}[k]=\sum_{i=0}^{N}\omega_{sen}^{i},$$


$J_{pers}[k]$ represents the persistent tracking penalty function value at time k. When the penalty function value is 0, it means that all UUVs can observe the target. N represents the number of UUVs of the tracker and $\omega_{sen}^{i}$ is the persistent tracking penalty factor. The expression is as follows:(29)$$\omega_{sen}^{i}=\left\{\begin{array}{c} 0,X_{t}[k]\in \Omega_{sen}^{i}\\ 1,X_{t}[k]\notin \Omega_{sen}^{i} \end{array}\right.,$$
where i=1⋅⋅⋅N, N is the number of tracker UUVs, $X_{t}[k]$ is the position of the target at time k, $\Omega_{sen}^{i}$ is the sensor detection range of the i-th UUV, $X_{t}[k]\in \Omega_{sen}^{i}$ means that the target at time k is within the sensor detection range of the i-th UUV, and $X_{t}[k]\notin \Omega_{sen}^{i}$ means that the target at time k is not within the sensor detection range of the i-th UUV.

(2)Path-crossing penalty function


(30)
$$J_{cros}(X_{com}[k],X_{uuv}[k])=\sum_{i=0,j=0}^{N}\omega_{cros},(i\neq j),$$


In the formula, $X_{uuv}[k]$ represents the position of the UUV at time k, $X_{com}[k]$ represents the instruction position of the future UUV optimized by the particle swarm optimization algorithm at time k, N represents the number of tracker UUVs, and $\omega_{cros}$ is the path-crossing penalty factor, specifically expressed as:(31)$$\omega_{cros}=\left\{\begin{array}{c} 0,\;Route_{i}\cap Route_{j}=\emptyset \\ 1,\;Route_{i}\cap Route_{j}\neq \emptyset \end{array}\right.,$$
where i,j=1⋅⋅⋅N and i≠j, $Route_{i}$, and $Route_{j}$ are the paths of the i-th UUV and j-th UUV, respectively. $Route_{i}\cap Route_{j}=\emptyset$ means that the two UUV paths have no crossing and $Route_{i}\cap Route_{j}\neq \emptyset$ means that the paths of the two UUVs have crossings.

As for the specific judgment method for path crossing, the current position of the UUVs, and the corresponding instruction position of the particle swarm optimization income as a path segment, there are several trackers with several paths according to the quick rejection in mathematical experiments, judging the line segment intersection method to determine whether the paths intersect, and then judging whether the instruction of the particle swarm optimization algorithm is reasonable. The specific path-crossing judgment method process is described as follows:

Firstly, the current position of the UUV is taken as the starting point of the path segment and the corresponding instruction position of the UUV is taken as the end point of the path segment. Each path segment is represented by the starting point and the end point. Then, the path segment of the i-th UUV can be represented as $Route_{i}=\{X_{i}[k],X_{com}^{i}[k]\}$, so that there are several UUVs and path segments. Observe the cross-product of four points.

(3)Instruction safety penalty function


(32)
$$J_{safe}(X_{t}[k],X_{com}[k])=\sum \omega_{safe},$$


In the formula, $X_{t}[k]$ is the position of the target at time k, $X_{uuv}[k]$ represents the position of the UUV at time k, and $\omega_{safe}$ is the instruction safety penalty factor, as detailed below:(33)$$\omega_{safe}=\left\{\begin{array}{c} 0,\\ 1, \end{array}\begin{array}{c} X_{com}[k]\in \Omega_{safe}\\ X_{com}[k]\notin \Omega_{safe} \end{array}\right.,$$
where $\Omega_{safe}$ represents the safe area. The specific meaning of the safe area is that the safe distance should be kept between the UUV instructions, the safe distance should be kept between the UUV instructions and the target position, and the instruction should not be within the prohibited navigation area. $X_{com}[k]\in \Omega_{safe}$ indicates that the instruction position of the optimized UUV meets the safety requirements and $X_{com}[k]\notin \Omega_{safe}$ indicates that the optimized instruction position does not meet the safety requirements.

### 5.2. Cooperative Optimization Algorithm for Tracking Path Based on PSO

Particle swarm optimization (PSO) is a swarm intelligence optimization algorithm based on birds searching for food [17]. The core of the particle swarm optimization algorithm is two formulas, namely, the velocity update formula and the position update formula:(34)$$V_{iD}^{k+1}=\omega \cdot V_{iD}^{k}+c_{1}r_{1}(P_{i}-X_{iD}^{k})+c_{2}r_{2}(P_{g}-X_{iD}^{k}),$$
(35)$$X_{iD}^{k+1}=X_{iD}^{k}+V_{iD}^{k+1},$$
where $V_{iD}^{k+1}$ is the D-dimensional velocity of the particle at time k+1, $X_{iD}^{k+1}$ is the D-dimensional position of the particle at time k+1, ω is the inertia weight, $c_{1}$ and $c_{2}$ are the learning factors, and $r_{1}$ and $r_{2}$ are random numbers. In practice, the following Equation (33) is often used to update ω [18]:(36)$$\omega =(\omega_{max}-\omega_{min})\frac{iter_{max}-iter}{iter}+\omega_{min},$$

In the formula, $iter_{max}$ is the maximum iteration number of the algorithm, iter is the current iteration number, $\omega_{max}$ is the initial inertia weight, $\omega_{min}$ is the minimum inertia weight, and the value is generally $\omega_{max}=0.9$, $\omega_{min}=0.4$.

In this paper, combined with the tracking path optimization problem of three UUVs’ cooperative target tracking, the three UUVs are regarded as a particle, and each particle contains the two-dimensional coordinates of the three UUVs, so the particle dimension is two-dimensional. The optimal solution of the optimization problem is the two-dimensional coordinates of the three UUVs that meet the task requirements. In each iteration, each particle is evaluated by the fitness function designed in the previous section, and the particles are updated by Equations (34)–(36). Finally, the optimal solution is obtained or the minimum accuracy requirement is achieved through layer-after-layer iteration.

The detailed steps of the multi-UUV cooperative target tracking path optimization algorithm based on particle swarm optimization are shown in Table 2:

In this table, $X_{u}[k]$ is the current position of the UUV, N is for tracking the target number of UUVs, $X_{t}^{-}[k]$ is for the prediction of the target position, fit(i) is the i-th particle fitness value of the current position, partile(i).position is the current position of the i-th particle, partile(i).positionBest is the best position of the i-th particle, partile(i).fitBest is the fitness value of the best position of the i-th particle, population.positionBest denotes the best position of the population, population.fitBest denotes the fitness value of the best position of the population, partile(i).V is the current velocity of the particle, iter denotes the current iteration number of the algorithm, $iter_{max}$ is the maximum number of iterations of the algorithm, Δ is the accuracy of the current solution, and $\Delta_{min}$ is the particle swarm optimization accuracy threshold. When the accuracy requirements are met, it is considered that the optimal solution of the problem has been found and the iteration can be ended. In this paper, according to the characteristics of the fitness function designed in Section 5.1, when the optimal fitness value of the population is unchanged for N consecutive beats, it is considered that the requirements have been met.

### 5.3. Target State Prediction Methods

In Section 3.1 of the paper, the closed-loop target tracking task process with two different path optimization bases was discussed. In the process of tracking path optimization, if the algorithm uses a long time or the UUV communication takes a long time, the path optimization based on the current target fusion estimation results may be at risk of target loss. It is more reasonable to predict the state of the target in the future based on the estimated state of the target fusion and use the prediction results as the basis for path optimization, so the method can greatly reduce the probability of target loss. The specific prediction method is as follows.

Firstly, given the number of models j in each IMM-EKF filter and the initial probability $\mu_{i}[0]$ of the model, after each filtering, the model probability $\mu_{i}[k]$ is updated. According to $\mu_{i}[k]$, the motion model $F_{j}[k]$ that most conforms to the real motion state of the target at the current time k is judged, and the target prediction state is:(37)$$X_{t}^{-}[k+T_{fo}]=F_{j}[k]\cdot X_{t}[k],$$

$F_{j}[k]$ is the target model with the maximum corresponding probability $\mu_{ij}$ in the IMM-EKF algorithm, $X_{t}[k]$ is the fusion estimation state of the target at time k, and $T_{fo}$ is the prediction duration.

## 6. Simulations

In order to verify the feasibility of the method, this section combines the algorithms in Table 1 and Table 2 to simulate and verify the whole process of multi-UUV cooperative target tracking, and the specific simulation design is described below.

(1)The total simulation time was 1500 beats and the sampling time was T=0.5 s.(2)Target setting. The target state variable was set to $X_{t}={\left[\begin{array}{cccc} x_{t} & y_{t} & \overset{•}{x_{t}} & \overset{•}{y_{t}} \end{array}\right]}^{T}$ and the CV model and CT model established in Section 2.3 were used for the target motion model. In the simulation process, the target performed a turning motion with an angular velocity π/600(rad/s) from 150 s to 225 s, a turning motion with an angular velocity −π/500(rad/s) from 375 s to 400 s, a turning motion with an angular velocity −π/800(rad/s) from 525 s to 600 s, a turning motion with an angular velocity π/900(rad/s) from 600 s to 675 s, and a straight-line motion with uniform speed in the rest of the time. The initial state variable of the target was set to $X_{t}[0]=[0m,2.5\mathrm{m/s},0m,2.3\mathrm{m/s}]$, and the motion plane was two-dimensional.(3)UUV setting. In the simulation, using three UUVs for target tracking, a PID controller was used to achieve UUV movement control, using a dynamic model for the UUV heading and X, Y directional thrust. Each UUV carries passive sonar, which can only detect the target’s bearing; the detection radius was set to 300 m, the detection azimuth was Z[k], and the detection noise was Gaussian white noise with a mean of 0 and variance of ${\left(0.01\mathrm{rad}\right)}^{2}$. The UUVs can communicate with each other by underwater sound with a communication radius of 300 m.(4)The settings of the distributed cooperative target state fusion estimation algorithm based on IMM-EKF. Each UUV used the IMM-EKF algorithm to estimate the target state. Additionally, UUVs can be federally fused locally, according to the target information sent by other UUVs and the local target information. Among them, IMM-EKF contained CV and CT models, the initial model probability was μ[0]=[0.5,0.5], and the Markov probability transition matrix of the system was set as follows:


$$P_{ij}=\left[\begin{array}{cc} 0.99 & 0.01\\ 0.01 & 0.99 \end{array}\right],$$


The system noise for all three filters was set as follows:$$Q[k]=0.01^{2}*diag([1m,1m,1\mathrm{m/s},1\mathrm{m/s}]).$$

(5)The settings of the collaborative optimization algorithm for tracking a path based on particle swarm optimization. The particle swarm size was set to 200 particles, the maximum number of iterations was 1000, the learning factors were $c_{1}=2.8$ and $c_{2}=1.3$, and the inertia weight was updated using Equation (36); the initial inertia weight was 0.9, and the minimum inertia weight was 0.4. The iteration was terminated when the optimal fitness of the population was unchanged for 100 consecutive beats.

Firstly, the current target fusion estimation state was used as the basis for path optimization, and the simulation results were as follows:

In the zoom-in position of Figure 7, it can be seen that UUV1 lost the target, which did not meet the task requirement of persistent tracking.

Then, the above parameters remained the same. Taking the target prediction state as the input basis for path optimization, the prediction time was set to half of the particle swarm optimization time, and the simulation result is shown as follows.

According to the analysis of the multi-UUV cooperative target tracking simulation diagram in Figure 8, the overall navigation path of the UUV was consistent with the target navigation path. It can be seen from the local enlarged image that the target was within the detection range of the three UUVs in the linear motion and the turning motion, which met the task requirements of persistent tracking, and the UUV navigated towards the command position optimized by the particle swarm at each time.

Figure 9 and Figure 10 show the estimated results and estimated root mean square error (RMSE) of the target position. It can be seen that the IMM-EKF algorithm could also obtain good estimation results for a single UUV when the target moved in a uniform straight line. However, when the target moved in a turn, the RMSE of the single UUV estimation results fluctuated greatly, and the estimation results were not ideal. However, the RMSE of the three UUV estimation results was significantly reduced after using the federal fusion filtering algorithm. This shows that the IMM-EKF algorithm could obtain better motion state estimation results when the target moved in a uniform straight line, but when the target turned, the single UUV estimation results fluctuated greatly. The federal fusion could make up for this defect. It could also obtain good estimation results when the target turned and moved.

Figure 11 shows the real-time distance between each UUV and the target. Figure 12 shows the real-time distance between the UUVs. As can be seen in Figure 12, there would be no collisions between the UUVs in the tracking process (due to the large range of distance variation between the UUVs, the coordinates of the points with small values are marked in the figure to show that there was no collision, even when the distance between the UUVs was small), which indicates that the fitness function designed in Section 5.1 played the effect of collision avoidance between the UUVs. Figure 11 shows that the distance between each UUV and the target during the tracking process was always less than 300 m, which realizes the purpose of persistent tracking.

## 7. Conclusions

The distributed cooperative target state fusion estimation algorithm based on IMM-EKF proposed in this paper can estimate target motion state according to bearing-only information well, obtain a smooth estimation result in a target’s straight motion and turning motion, and has a high estimation accuracy. The cooperative path optimization method based on the particle swarm optimization algorithm can obtain the path points that meet the requirements of persistent tracking according to the target fusion estimation results. The optimization function designed in this study not only completed the task requirements of persistent tracking, but also had a beneficial effect on UUV collision avoidance, such as the path-crossing penalty factor and instruction safety penalty factor. The simulation results show that the distributed cooperative target state fusion estimation algorithm based on IMM-EKF and cooperative path optimization method based on particle swarm optimization can ensure that multiple UUVs can track a target without losing it and avoid collisions when the target moves in a uniform straight line or a turning direction.

## Figures and Tables

**Figure 1 sensors-23-07865-f001:**
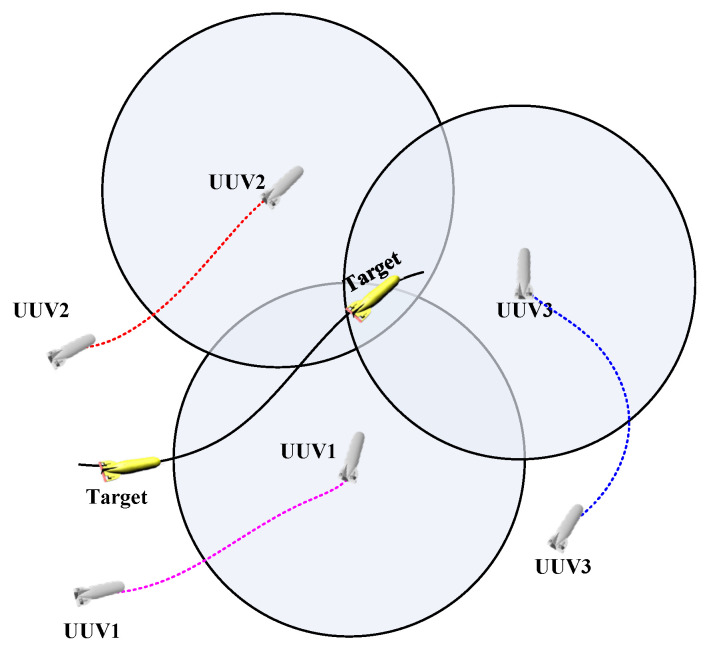
Schematic diagram of multi-UUV cooperative target tracking.

**Figure 2 sensors-23-07865-f002:**
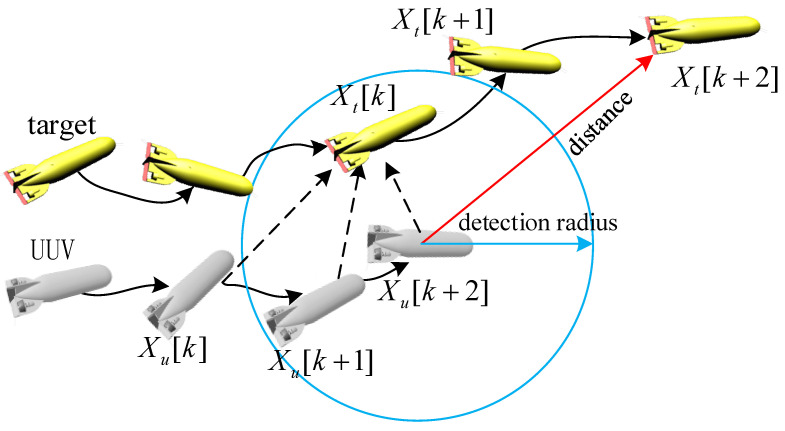
Tracking optimization approach based on target current states.

**Figure 3 sensors-23-07865-f003:**
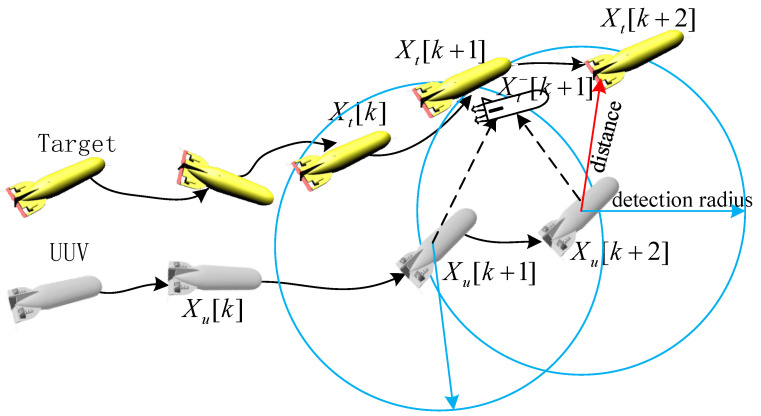
Tracking optimization approach based on target prediction states.

**Figure 4 sensors-23-07865-f004:**
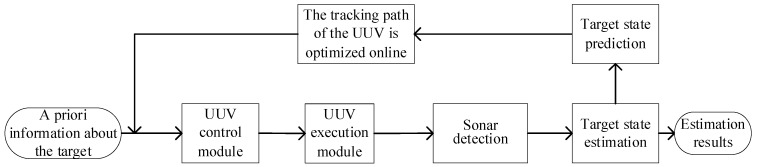
Tracking structure based on target predicted state.

**Figure 5 sensors-23-07865-f005:**
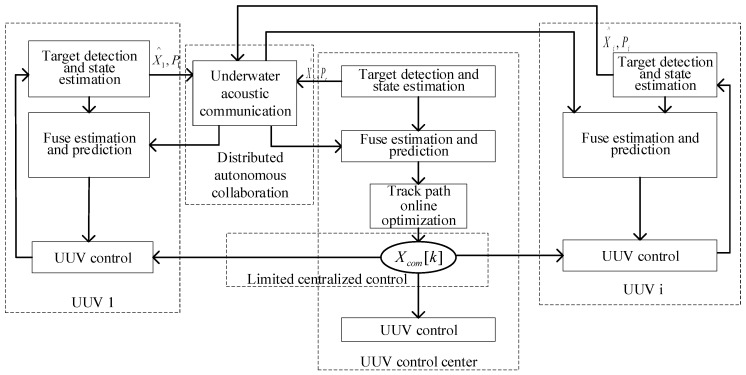
Finite centralized distributed solving framework.

**Figure 6 sensors-23-07865-f006:**
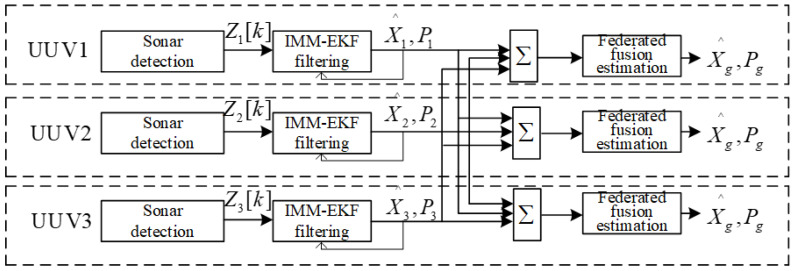
Structure of distributed fusion estimation based on IMM-EKF and federated fusion.

**Figure 7 sensors-23-07865-f007:**
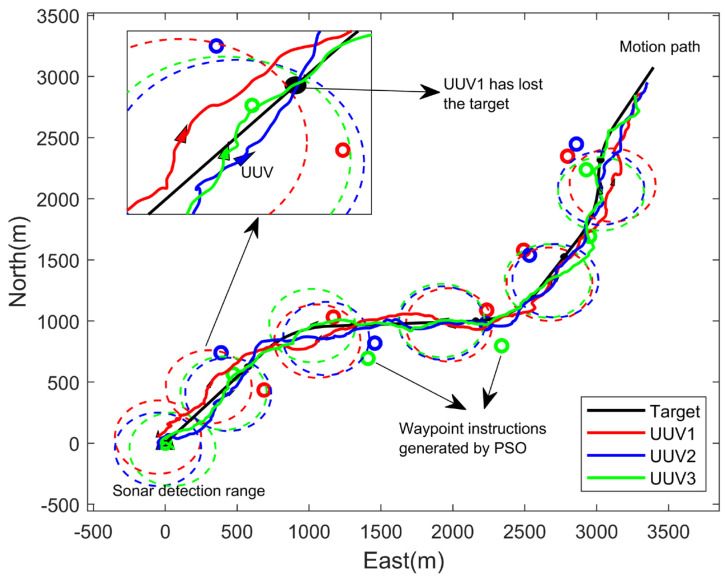
Cooperative target tracking result of three UUVs (optimized based on current state).

**Figure 8 sensors-23-07865-f008:**
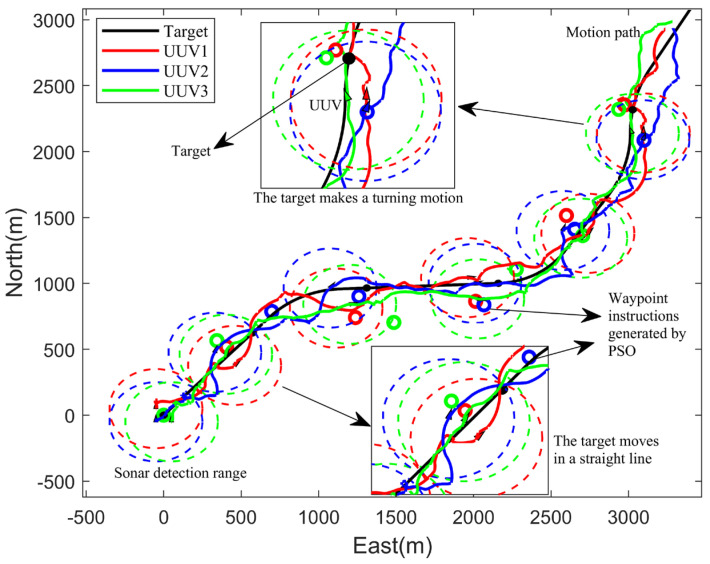
Cooperative target tracking result of three UUVs.

**Figure 9 sensors-23-07865-f009:**
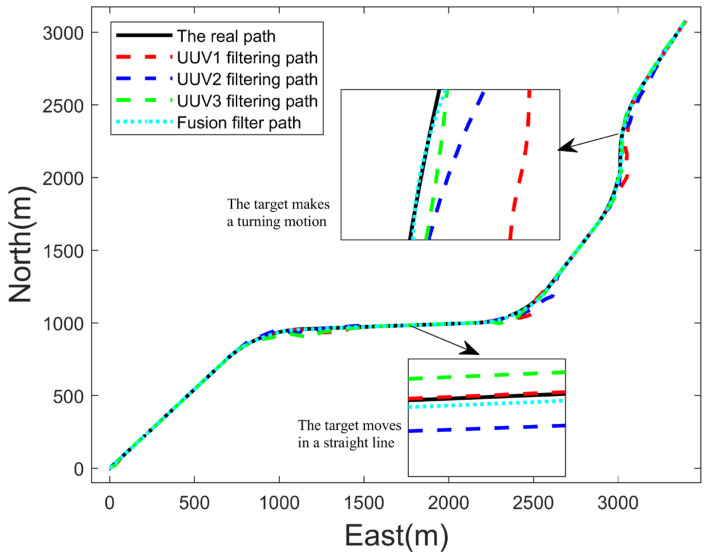
Estimated result of target position.

**Figure 10 sensors-23-07865-f010:**
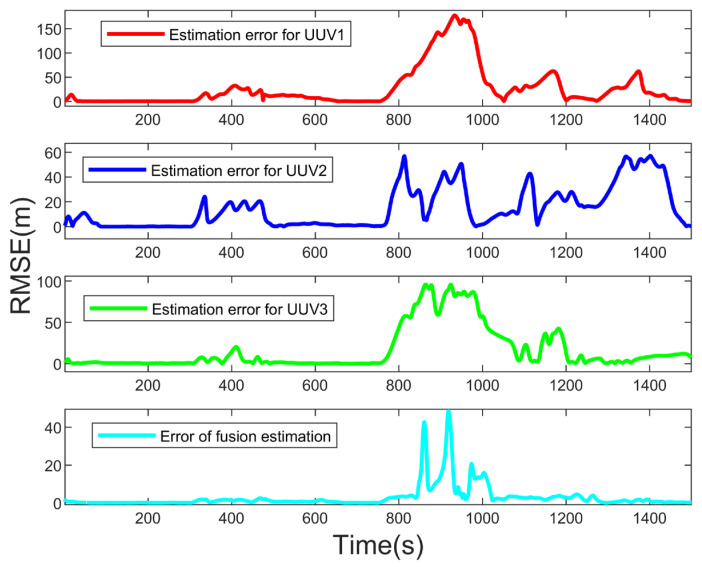
Estimated root mean square errors of target position.

**Figure 11 sensors-23-07865-f011:**
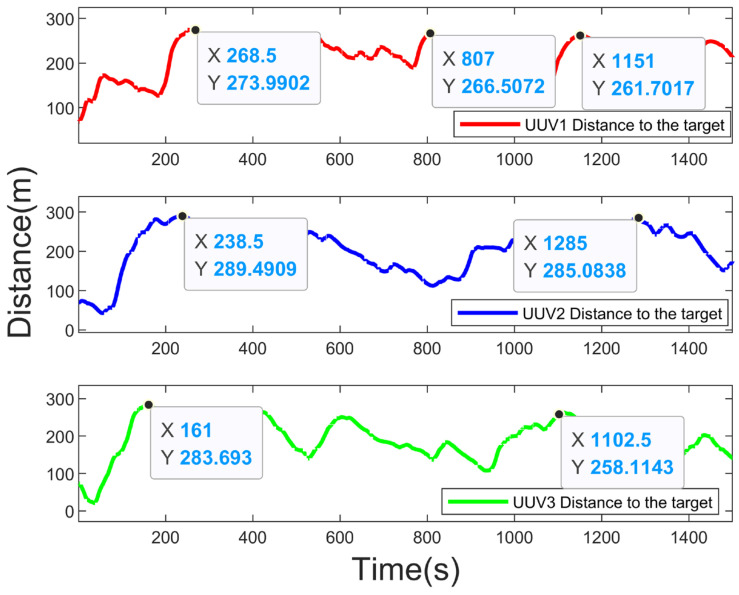
Real-time distance between each UUV and target.

**Figure 12 sensors-23-07865-f012:**
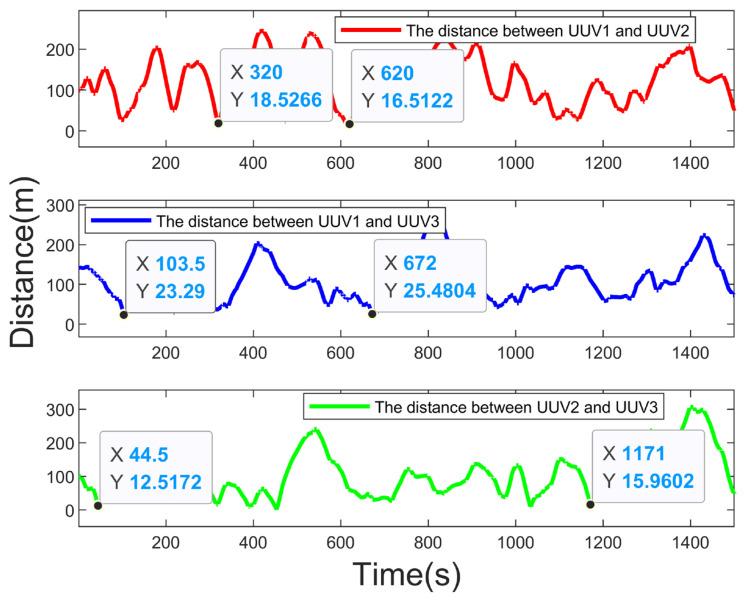
Real-time distance among UUVs.

**Table 1 sensors-23-07865-t001:** Distributed cooperative target state fusion estimation algorithm based on IMM-EKF.

Step 1 Initialization. Set each UUV $\hat{X_{i}}[0]$, $P_{i}[0]$, $p_{ij}$, $\mu_{i}[0]$, i=1,⋅⋅⋅,N. Step 2 IMM-EKF. Each AUV is executed: Step 2.1 Input interaction. Use Equations (8)–(11) to obtain the mixed input under each target model. Step 2.2 EKF. Use the mixed input obtained in the previous step to obtain the estimation results under each model according to Equations (12)–(17). Step 2.3 Model probability update. Obtain the model probability at the current moment $\mu_{i}[k]$. Step 2.4 Mixed output. The local estimation results of each UUV are obtained using Equations (23) and (24) ${\hat{X}}_{i}(k|k)$ and $P_{i}(k|k)$. Step 3 UUV communication. Each UUV sends ${\hat{X}}_{i}(k|k)$ and $P_{i}(k|k)$ to the other UUVs through underwater acoustic communication. Step 4 Fusion estimation. Each UUV obtains the final target state estimation results $\hat{X_{g}}$ and $P_{g}$ through Equations (25) and (26). Step 5 Repeat Step 2–Step 4 until the task is over.

**Table 2 sensors-23-07865-t002:** Cooperative optimization algorithm of tracking path based on PSO.

Step 1: Input $X_{u}[k]$, N, $X_{t}^{-}[k]$. Step 2: Initialization population. Set $c_{1}$, $c_{2}$, ω, $V_{max}$; Set particle & population; Step 3: $fit(i)=J_{pers}+J_{cros}+J_{safe}$ Step 4: if (fit(i)≤partile(i).fitBest) partile(i).positionBest=partile(i).position if (fit(i)≤population.fitBest) population.positionBest=partile(i).position end end Step 5: Update partile(i).V &partile(i).position according to Equations (34) and (35); Step 6: if ($iter=iter_{max}$ or $\Delta \leq \Delta_{min}$) Step 7; else Step 3; end Step 7: Output $X_{com}$. Step 8: UUV execute $X_{com}$.

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
