# Peer review of "Research on the Cooperative Target State Estimation and Tracking Optimization Method of Multi-UUV"

_sensors, 2023, doi:10.3390/s23187865_

Round 1

Reviewer 1 Report

This paper studies the problems of cooperative target state estimation and tracking optimization for multi-UUV target tracking problem. It develops an IMM-EKF to estimate the target state, and then uses the particle swarm optimization to online collaboratively optimize the UUV tracking path. The paper is organized reasonably and the results are convinced. However, it still suffers from some major issues to be addressed:

1. The language description of this paper is very poor, the authors should check the English grammar and language quality carefully;

2. The literature citation has some errors. Please check.

3. The full name of UUV should be given;

4. The literature review is limited, the research status on cooperative target state estimation should be provided. Please check “Mahalanobis distance-based fading cubature Kalman filter with augmented mechanism for hypersonic vehicle INS/CNS autonomous integration”.

5. The IMM-EKF and particle swarm optimization are both the well-known approaches. Thus, in my viewpoint, the novelty of this paper is very limited. The novelty and contribution of this paper should be further stressed and stated.

6. The unit of RMSE in Fig.9 is missing;

7. The estimation accuracy of proposed method should also be compared with the existing approaches, for example

Overall, this manuscript can be accepted by the Journal “Sensors” after major revision.

The language description of this paper is very poor, the authors should check the English grammar and language quality carefully.

Reviewer 2 Report

The paper proposes a solution to two sub-problems in multi-UUV cooperative target tracking: cooperative state estimation and optimization of tracking path. The authors establish a mathematical model of the system and use the extended Kalman filter algorithm and federal fusion algorithm for distributed target state estimation. They also use the particle swarm optimization algorithm for online collaborative optimization of UUV tracking path. Simulation results demonstrate the effectiveness of the proposed approach in achieving better target state estimation and persistent tracking of the target. However, it needs to address the following issues:

1.       Many reference sours are not founded, please reformat the reference citations.

2.       For Eq. (5), the representation of the terms ‘i’ and ‘j’ should be introduced before Eq. (5).

3.       In line 262, the authors firstly announced the local filters, while the expression of local filter is not clear in this case. It would be more appropriate to use ‘sub-filters’.

4.       In line 117 and 120, indeed, the prospect of target loss remains a possibility when only the present states are taken into consideration. Nevertheless, a multitude of studies have addressed this predicament through a diverse range of techniques, such as random weighting, Sage windowing, and adaptive factor adjustments. As such, an exploration and analysis of the aforementioned improvements for resolving the aforementioned issue would be greatly appreciated. Please check the papers:

[1] Sage windowing and random weighting adaptive filtering method for kinematic model error

[2] Interacting multiple model estimation-based adaptive robust unscented Kalman filter

5.       The computational burden of the proposed method should be evaluated by comparing with the traditional method.

6.       According to the ‘Crossref’, this paper is with high similarity to the article named “Estimation of Vehicle State Based on IMM-AUKF” and “Interacting multiple model estimation-based adaptive robust unscented Kalman filter”. The authors should discuss the comparative analysis refer to the articles.

7.       The highlight of this paper is about using the fusion estimation techniques in EKF, but barely discussed the advantages about EKF, the authors should comprehensively introduce the EKF and its superiorities. Please refer to:

[2] Extended Kalman filter based on stochastic epidemiological model for COVID-19 modelling

[3] Extended Kalman filter for online soft tissue characterization based on Hunt-Crossley contact model

Overall, this manuscript can be accepted by the Journal “Sensors” after major revision.

The text may benefit from some improvements to make it more readable and expressive. There are some grammar errors and the sentences are quite prolix, which can make it difficult to understand. With some effort, the author could enhance the clarity and coherence of their writing.

Reviewer 3 Report

This manuscript studied multi-UUV cooperative target tracking problem, a distributed target state estimation algorithm based on IMM-EKF is proposed to solve the problem of multi-UUV cooperative target tracking. The extended Kalman filter, federated fusion and prediction algorithm are used to improve the accuracy of target state. In addition, the particle swarm optimization algorithm is used to online collaborative optimization of the UUV tracking path.

This manuscript still has the following problems:

1. The presentation of the current research on cooperative target tracking problem is too brief to provide a comprehensive understanding of the difficulties of current research to be addressed. The references are old and few in number, and in particular the results of progress in the last two years are not presented.

2. The innovation of the manuscript is not clear enough. There is no comparison with other methods, and innovation is difficult to assess.

3. The manuscript has a lot of formatting errors, including graphs, symbols, formulas, literature citations, etc. It is poorly readable.

4. The language of the manuscript needs to be improved.

5. The Definition 1.1 Multi-UUV cooperative target tracking problem should be mathematical modeling.

6. The parameters used in the algorithm were not specified during the simulation process.

7. What are the communication requirements between UUVs in order to achieve collaboration?

8. Is Interacting Multiple Model (IMM) self proposed or cited?

The language of the manuscript needs to be improved.

Round 2

Reviewer 1 Report

1. There are some blank in Fig.6 left behind, please fill.

2.  The authors note ∆min as the particle swarm optimization accuracy threshold, please explain how the threshold is determined.

The language had been improved.

Reviewer 2 Report

i)         There is some missing information in textbox of Fig.6.

ii)       The authors still need to provide full description for given abbreviations, e.g. ‘CKF’ and ‘HV’ in line 36-39.

The language is well improved

Reviewer 3 Report

The modified part is worthy of recognition. Further grammar modifications are needed.

The modified part is worthy of recognition. Further grammar modifications are needed.
